# High-Intensity Laser Therapy in Pain Management of Knee Osteoarthritis

**DOI:** 10.3390/biomedicines12081679

**Published:** 2024-07-27

**Authors:** Daniela Poenaru, Miruna Ioana Sandulescu, Claudia Gabriela Potcovaru, Delia Cinteza

**Affiliations:** Rehabilitation Department, Carol Davila University of Medicine and Pharmacy, 050474 Bucharest, Romania; miruna.sandulescu@drd.umfcd.ro (M.I.S.); claudia-gabriela.potcovaru@drd.umfcd.ro (C.G.P.); delia.cinteza@umfcd.ro (D.C.)

**Keywords:** knee osteoarthritis, high-intensity laser therapy, pain management, physical stimulation

## Abstract

Knee osteoarthritis (KO) is an important health condition, affecting one third of people aged 65 years or more. Pain is the main cause of disability. Pain management in KO includes pharmacological and non-pharmacological modalities. Patient education, lifestyle changes, physical exercise, and physical agents are prescribed as a first approach for pain control. Laser therapy is part of many therapeutical protocols, with two forms: low-level laser therapy (LLLT) and high-intensity laser therapy (HILT). This paper aimed to stress the advantages of HILT based on a greater wavelength, higher energy delivery, and deeper tissue penetration. Research on 23 published trials revealed that the analgesic effect is rapid, cumulative, and long lasting. Compared to sham, to LLLT, or to other combinations of therapeutical modalities, HILT provided significantly better results on pain reduction and functional improvement. Ultrasound examination showed a reduction in intra-articular inflammation.

## 1. Introduction

Knee osteoarthritis (KO) represents an important health issue, affecting one third of people over the age of 65 years [1]. Before the age of 50 years, men are more prone to develop KO, and after this age, women are more affected, as more complex hormonal and metabolic changes occur [2]. Furthermore, people of younger ages are prone to develop joint disease, especially KO, as a consequence of obesity, despite the combined efforts of various approaches to limit it [3]. KO disability arises mainly from pain and, as disease progresses, from movement limitation, misalignment, and muscle imbalance. 

Laser therapy (LT) is a non-invasive and painless process which can be applied in a wide variety of conditions [4]. Although low-level laser therapy (LLLT) is more commonly used, high-intensity laser therapy (HILT) is a new, painless, and strong alternative with higher density radiation. With a wavelength of 1064, HILT devices can provide deeper penetration. As a difference from LLLT, the biological effects of HILT are based less on chromophore absorption and rather on photochemical reactions (as increasing mitochondrial oxidative reaction and adenosine triphosphate, RNA, and DNA production) and photobiostimulatory effects. The advantage of HILT over LLLT is that it can reach deeper and larger joints. Through deep thermal action, HILT increases local blood circulation, improves tissue regeneration, and reduces pain and edema. A few studies conducted on osteoarthritis, particularly on knee osteoarthritis (KO) have drawn attention to this topic, and we aimed to summarize their findings in this paper to draw attention to a new therapeutical method.

Miofascial pain syndrome is a very common cause of disability in the population and an important issue to be addressed in therapy. HILT associated with physical exercise was found to be more effective that exercise alone in trapezius localization [5,6]. HILT was effective in improving pain in cervical myofascial pain syndrome but not the cervical range of motion (ROM), concluded a meta-analysis, mentioning that due to the heterogeneity of the trials, the latter allegation could be biased [7]. For chronic neck pain, an association of HILT and physical exercise significantly improved pain and mobility [8]. Subacromial impingement syndrome was successfully managed with a combination of HILT and exercise in terms of pain and function [9]. Patients with low back pain, a frequent health issue, reported an improvement in pain and functional status when receiving HILT and physical exercise [10].

Controversial results were reported for HILT in plantar fasciitis management, as different groups of researchers stressed either the effectiveness [11] or the lack of it [12,13] in pain reduction or in functioning [14], inviting further studies. Furthermore, a systematic review from 2020 concluded that HILT may be beneficial in a polymodal approach to musculoskeletal pain [15].

## 2. Materials and Methods

Two researchers (D.P. and M.I.S.) independently performed a search of the available literature, from inception to May 2024, from databases such as PubMed, Google Scholar, and Web of Science, using a combination of MeSH terms: “high-intensity laser therapy”, “high-level laser therapy”, “high-power laser”, “HILT”, “HLLT”, “knee osteoarthritis”, and “gonarthrosis”.

Following the PICO model, we defined the inclusion criteria. The inclusion criteria were as follows: (1) papers containing randomized controlled trials, case reports, case series, or pilot studies on human adult subjects (above 18 years old) with a (2) diagnosis of KO based on ACR criteria (P); (3) HILT is referenced as a therapeutical intervention (I), (4) compared to sham, LLLT, and other therapeutical interventions (C); and there are (5) well-defined clinical outcomes: improvement in pain, function, disability, ultrasound examination (O). We selected articles written in English with the full text available. The selection offered 136 papers. Duplicates were excluded, and 115 records remained. During the eligibility process, the authors excluded reviews, systematic meta-analyses, studies on animal subjects, and in vitro studies, and 24 papers remained. One paper lacked full text in English and was excluded. Finally, 23 papers fulfilled the criteria and were included in the actual research. Disagreements were discussed until a consensus was reached. When needed, a third researcher (D.C.) became involved. According to the PEDro scale on the quality of RCT, one study had a score under 5; however, the authors decided to retain the paper upon consideration of the number of patients and the accuracy of the protocol. Another trial scored 6 on the PEDro scale, and the remaining 21 papers scored between 7 and 11. A score of ≥7 is considered to be a study of high methodological quality, while a score of ≤5 is considered to be of low methodological quality. 

See Figure 1 for the selection process of relevant trials.

## 3. Results

The selected 23 papers (with a total number of 1175 patients) were as follows: one prospective, clinical, before and after treatment [16]; one prospective, clinical, before and after treatment case series [17]; one prospective, open label, before and after clinical trial [18]; and twenty prospective randomized trials [19,20,21,22,23,24,25,26,27,28,29,30,31,32,33,34,35,36,37,38].

As specified in the inclusion criteria, all papers included patients with KO according to ACR criteria. The severity of KO was assessed with Kelgren and Lawrence criteria, and all papers included patients with grades 2 and 3. Pain was the main outcome for all trials and was assessed with the visual analogue scale. Function was noted mostly with the WOMAC index and range of motion (ROM), especially with respect to active flexion; one trial used also the Lequesne scale [18]. Disability was assessed in two studies with the Knee Injury and Osteoarthritis Outcome Score (KOOS), an auto-administered questionnaire [34,35]. Gait analysis was performed in a small number of studies using different tools (pedobarometry, walking distance without pain, timed up and go test (TUG), 6-min walk test (6 MWT), 50-feet walk test). Four studies performed knee ultrasound examination following cartilage and synovial thickness and intra-articular fluid [24,27,32,37]. One trial used the Berg balance scale [38]. See Table 1 for the selected trials included in the research.

**Table 1 biomedicines-12-01679-t001:** The selected trials included in this research.

Author, Year of the Trial	Trial Type, No of Patients	Intervention	Outcome	Moments of the Study	Results
Viliani, 2009 [19]	Prospective, randomized, 41 pts	Group A: HA intra-articular (4 weekly infiltrations). Group B: HILT, 10 sessions on alternate day.	Function (WOMAC; Lequesne scale)	Baseline End of the treatment 4 months	Intragroup: significant improvement at the end of the treatment, maintaining the values at 4 months
Sabbahi, 2009 [20]	Prospective, randomized, 30 pts	HILT + ET. LLLT + ET. US + ET. 2 sessions/week, 3 weeks	Pain (VAS), Walking distance without pain (in meters), Squatting with/without pain.	Baseline End of the treatment	Intragroup: improvement for all US and LLLT equally efficient Significantly better results for HILT
Stiglig-Rogoznica, 2011 [16]	Prospective, clinical, before and after treatment, 96 pts	HILT, 10 consecutive days	Pain (VAS)	Baseline End of the treatment	Pain significantly decreased
Viliani, 2012 [18]	Prospective, open label, before–after, clinical trial, 34 pts (41 knees)	Group A: HILT (10 sessions, 3000 J/session, on alternate days) Group B: control	Function (WOMAC)	Baseline End of the treatment 4 months	Group A improved significantly after treatment and at 4 months (with a slight regression); Group B showed worsening of the parameter
Kheshie, 2014 [21]	Prospective, single-blinded randomized controlled trial, 53 pts	Exercise + HILT Exercise + LLLT Exercise + sham 2 sessions/week, ×6 weeks	Pain (VAS) Function (WOMAC)	Baseline End of the treatment	Both LLLT and HILT were better than sham HILT improved significantly better than LLLT
Kim, 2016 [22]	Prospective, randomized, 20 pts	Group CPT (conservative physical therapy), Group HILT (CPT + HILT) 3 times/week, 4 weeks, 1500 J/cm^2^	Pain (VAS) Function (WOMAC)	Baseline End of the treatment	Intra-group: both groups improved statistically Inter group: HILT had better scores
Angelova, 2016 [23]	Prospective, pilot trial, randomized, clinical, single-blinded, placebo controlled, 72 pts	HILT (300 J/session analgesic, 3000 J/session biostimulatory) versus sham, 7 daily sessions	Pain Pedobarometric gait analysis (static and dynamic)	Baseline End of the treatment 1 and 3 months	HILT group improved significantly after treatment and results were maintained at follow-up
Alayat, 2017 [24]	Prospective, single-blinded randomized controlled trial, 67 pts	Group 1: HILT, GCS, ET. Group 2: GCS + ET. Group 3: placebo + ET. 2 sessions/week, 6 weeks	Pain (VAS) Function (WOMAC) US (synovial thickness, femoral condylar cartilage)	Baseline End of the treatment 3 months	Intragroup: all have significant improvement at 6 weeks, persistent at 3 months, except US cartilage thickness Between groups: HILT improved better; no difference between groups 2 and 3
White, 2017 [17]	Prospective, case series, before and after treatment, 39 pts	1–3 daily sessions HILT	Pain (VAS) Function (ROM) Clinical issues (swelling, numbness)	Baseline End of treatment 1–3 weeks	Improvement of all items at all moments
Nazari, 2018 [25]	Prospective, assessor-blind, randomized controlled trial, 93 pts	HILT + ET, TENS + US + ET, ET alone, 12 sessions, 3 sessions/week	Pain (VAS) Function (flexion ROM; timed up and go test (TUG); 6-min walk test (6 MWT), WOMAC)	Baseline End of the treatment 12 weeks	Intragroup: improvements at all moments, all parameters Between groups: HILT had better results on all parameters
Ciplak, 2018 [26]	Prospective, randomized, single blinded, 48 pts	Hotpack + US + TENS + ET Hotpack + HILT + ET 10 sessions/2 weeks	Pain (VAS) Function (WOMAC)	Baseline End of the treatment 6 weeks	HILT was better significantly at all moments
Akaltun, 2021 [27]	Prospective, double-blind randomized placebo-controlled, 40 pts	HILT + ET (300 J/session analgesia, 3000 J/session biostimulation) Placebo + ET 5 sessions/week, 2 weeks	Pain (VAS) Functional (WOMAC; flexion ROM) US: cartilage thickness	Baseline End of the treatment 6 weeks	Both groups: 2 and 6 weeks: VAS, WOMAC—pain, WOMAC—function, WOMAC—stiffness, and WOMAC—total; cartilage thickness and FROM increased, 6 weeks: HILT + ET significantly better values for all parameters versus PL + ET
Koevska, 2021 [28]	Prospective, one-sided blind randomized comparative study, 72 pts	HILT versus LLLT 10 sessions	Pain (VAS)	Baseline End of the treatment 30 days	Both goups improved significantly After treatment, pain on VAS had lower scores for HILT groups that LLLT
Siriratna, 2022 [29]	Prospective, randomized, single-blind, parallel group study, 42 pts	Conservative treatment + HILT (562.5 J/session) Conservative treatment + sham 2–3 sessions/week, a total of 10 sessions	Pain (VAS) Function (WOMAC)	Baseline End of the treatment	Intra-group: both groups improved significantly in all items. Inter-group: HILT had lower pain scores (significant) No difference for WOMAC
Samaan, 2022 [30]	Prospective, single blinded, randomized, controlled trial 60 pts	HILT + ET, LIPUS + ET, ET alone, 5 sessions/week, 2 weeks	Pain (VAS) Function (ROM; WOMAC) Proprioception accuracy	Baseline End of the treatment	HILT better results in all parameters
Mostafa, 2022 [31]	Prospective, randomized controlled trial, 40 pts	ESWT, one session/week, 4 weeks. HILT, 3 sessions/week, 4 weeks	Pain (VAS) Function (6 MWT; WOMAC)	Baseline End of the treatment	Intragroup: both improved Between groups: HILT better results
Ekici, 2023 [32]	Prospective, double-blinded, placebo-controlled, randomized, 60 pts	Group 1 (HILT + hotpack + TENS + ET) 300 J/session followed by 3000 J/session Group 2 (sham laser + hotpack + TENS + ET) 9 sessions/3 weeks	Pain (VAS) Functional (flexion ROM, isokinetic muscle strength, WOMAC) US: cartilage thickness	Baseline End of the treatment 3 months	Both groups improved in all items at the end of treatment and at 3 months There was no difference between groups at any moment
Taheri, 2023 [33]	Prospective, randomized, controlled, 56 pts	ET + NSAID + topic ointment. ET + NSAID + topic ointment + HILT (3 session/week, 2 weeks)	Pain (VAS) Function (WOMAC)	Baseline End of treatment 3 months	All parameters were better in the HILT group at the end of the treatment and after 3 months
Katana, 2023 [34]	Prospective, descriptive, experimental, randomized trial, 60 pts	Group I, standard protocol + HIMS. Group II, standard protocol + HILT One session/week, 8 weeks	Pain (Likert scale) Functional (ROM) Disability (KOOS)	Baseline Middle (4 weeks) End of the treatment (8 weeks)	Intra-group analysis: both improved all parameters at all moments Intergroup analysis: Group II showed significantlu greater improvement in all moments
Ahmad, 2023 [35]	Prospective, randomized, double-blinded, parallel-group clinical trial, 34 pts	HILT + ET LLLT + ET Once a week, 12 weeks	Pain Disability (KOOS) Function (active flexion ROM; timed up and go test (TUG))	Baseline End of the treatment	Intragroup: all parameters improved HILT has significantly greater improvement
Astri, 2023 [36]	Prospective, double-blind randomized controlled clinical trial, 61 pts	LLLT + ET HILT + ET 3 sessions/week, 2 weeks	Pain Function (50-feet walk test)	Baseline Pain (after every session) End of the treatment	Pain improved in both groups, with better evolution at every moment for HILT Function improved better for HILT
Roheym, 2023 [37]	Prospective, randomized, double-blinded, pre/post-test trial, 30 pts with bilateral KO	ET ET + HILT (300 J/session, followed by 3000 J/session) 3 sessions/week, 4 weeks	US: suprapatellar fluid detection Function (WOMAC)	Baseline End of the treatment	Intragroup: significant improvements Better results in HILT
Wibisono, 2024 [38]	Prospective, randomized, pre-test and post-test-controlled, 27 pts	HILT versus LLLT 2 sessions/week, 4 weeks	Berg balance scale	Baseline End of the treatment	Intragroup: both groups improved significantly HILT improved

LLLT, low-level laser therapy; HILT, high-intensity laser therapy; HA, hyaluronic acid; ROM, range of motion; FROM, flexion ROM; VAS, visual analogue scale; HP, hot pack; ET, exercise therapy; HIMS, high induction electromagnetic stimulation; LIPUS, low-intensity pulsed ultrasound; KOOS, Knee Injury and Osteoarthritis Outcome Score; WOMAC, Western Ontario and McMaster Universities Osteoarthritis Index.

## 4. Discussion

The management of KO requires a complex approach, including medication, physical therapy, behavioral changes, and education. Intra-articular hyaluronic acid (HA) is an important step in this process. A prospective randomized study (2009) on 41 patients compared functional status after 4 weekly intra-articular infiltrations with HA versus 10 HILT sessions (2000–3000 J/session) on alternate days. Results were compared after therapy and 4 months later. Both groups improved significantly after treatment and maintained progress 4 months later, with no difference between the groups. This study introduced HILT as a tool in KO therapy [19]. 

A prospective, clinical, pre- and post-treatment trial (2011) on 96 patients treated with HILT, with ten consecutive daily sessions, showed significant improvement of knee pain after treatment [16]. A case series of 39 former football players with KO received one–three daily sessions of HILT and reported pain reduction and clinical improvement in swelling and numbness at the end of the treatment and maintenance of the result after 1 week and 3 weeks [17].

We present the papers according to the randomization method, comparing HILT to sham, to LLLT, and to other physical agents.

### 4.1. HILT versus Sham/Placebo

Eight trials with a total of 385 patients compared different regimens of HILT with sham (placebo). Two studies (123 patients) compared HILT alone with sham on pain, function, and gait [18,23]. They concluded that HILT was associated with significant improvements in all items after treatment, and the results were maintained at 3 and 4 months. The regimens were either 10 alternate-day sessions or 7 daily sessions, with 3000 J/session and a biostimulator effect. Six studies compared HILT associated with different conservative therapies (physical exercise; hot packs + interferential therapy + physical exercise; NSAID + topic ointments + physical exercise). Pain (assessed on VAS), function (WOMAC index), and ultrasonographic measurements (femoral condylar cartilage thickness and suprapatellar fluid) were performed at baseline, after treatment, and after a timeframe of 4 weeks to 3 months. All trials reported significant improvement in all parameters in the intragroup analysis for all moments. As for intergroup analysis, most trials (four out of six) reported significantly better results in the HILT group for pain and function. Ultrasound measurement of intra-articular fluid found an important reduction after HILT therapy [22,27,29,32,33,37]. One trial failed to identify a notable difference for HILT versus sham in pain and function [32], whereas one trial reported a significant improvement in pain only, not in function [29]. 

### 4.2. HILT versus LLLT

LLLT has been a part of pain therapy for a long time. Five trials compared the two forms of laser therapy (LLLT, HILT), either in monotherapy [28,38] or associated with physical exercise [21,35,36]. A total number of 247 patients reported significant improvements in pain (VAS scores), function (WOMAC index, 50-feet walk test, TUG), and disability (KOOS) in intragroup analysis at the end of therapy and one month later, with better results in the HILT group. Compared to sham, both LLLT and HILT were followed by clinical improvement. HILT produced significant improvement in pain scores versus LLLT in all trials. Function was documented in several ways, with an emphasis on disability evaluation (Berg balance scale, TUG, 50-feet walk test), and the results were significantly better for HILT patients. Protocols of HILT therapy varied from once a week for 12 weeks, two sessions/week for 4–6 weeks, three sessions/week for 2 weeks, to ten daily sessions. Doses were in the analgesic range (300 J) followed by a biostimulatory effect (3000 J) [21,28,35,36,38]

### 4.3. HILT versus Other Therapeutic Modalities

Six trials included 331 patients receiving therapeutic protocols using different forms of physical agents for analgesic purposes and therapeutical exercise to evaluate the effectiveness of HILT. It is worth noting that all the following schemes included physical exercise as a cornerstone, underlying its importance in the management of KO. The association with different modalities was thought to add value to therapy. Two sessions/week for 3 weeks of either HILT, LLLT, or therapeutical ultrasound showed significantly better results for HILT on knee pain and function (walking distance and squatting without pain) [20]. Three sessions/week for 4 weeks of either HILT or conventional therapy (TENS + ultrasound) were followed by significant improvements in pain and function (flexion ROM, TUG, 6 MWT, and WOMAC) in the HILT group. The difference was significant at 12 weeks after treatment completion, suggesting a persistent effect of HILT [25]. Ten HILT sessions over 2 weeks proved to be more effective on pain and function (ROM and WOMAC) than low-intensity pulsed ultrasound (LIPUS) [30]. HILT and hot pack for 10 sessions over 2 weeks proved to be more efficient than ultrasound, TENS, and hot pack on pain and function (WOMAC), with a lingering effect after 6 weeks [26]. HILT (three sessions/week, 4 weeks) was significantly more effective than ESWT (four weekly sessions) on pain and function (WOMAC, 6 MWT) after treatment completion [31]. Comparing eight weekly sessions of HILT to high-intensity magnetic stimulation (HIMS) with a standard protocol (shockwave therapy, TENS, massage, physical exercise) showed better scores for pain and function in the HILT patients after therapy [34]. Two of the six above-mentioned trials followed patients after the end of the therapy for 6 and 12 weeks and noted the persistence of significantly better results for HILT therapeutical schemes [25,26].

A study on different combinations of HILT, glucosamine/chondroitin sulphate (GCS), and physical exercise proved that two sessions/week for 6 weeks of HILT, together with GCS and physical exercise, led to significant improvements in terms of pain, function (WOMAC), and ultrasound-measured synovial thickness at the end of treatment, and the results were maintained 3 months later. Femoral condylar cartilage thickness remained unchanged during the study [24]. 

## 5. Conclusions

The analysis performed by the authors led to the conclusion that HILT is effective in pain reduction, outperforming LLLT and some combinations of physical modalities or drug administration. When function and disability were assessed, HILT may produce significant improvement. The success of the achievements was documented for an interval between one week and 4 months in 10 out of the 23 trials. 

Pain management in musculoskeletal pathologies, including osteoarthritis, is a major cause of disability and a challenge for the clinician. In the pharmacological approach, aside from conventional drugs (NSAIDs, analgesic), new modalities are investigated, such as platelet-rich plasma and botulinum toxin [38,39,40]. In the non-pharmacological approach, new forms of magnetic, ultrasound, shockwave, and laser therapy promise additional benefits [41].

The actual use of light therapy in rehabilitation includes lasers with two classes of power: class 3b, low-level laser therapy (LLLT) (500 mW or less); and class 4, high-intensity laser therapy (HILT) (greater than 500 mW.) LLLT, known under the name of cold lasers, with wavelengths between 632.8 nm and 830 nm, has a small penetration (2–3 mm to 2–4 cm) and produces biological effects after photon absorption by chromophores. HILT is defined by greater wavelength (1064 nm), which ensures a deeper penetration (up to 10–12 cm) and a greater energy delivery. HILT’s biological effects are based on thermal and chemical effects. The main indication of HILT is pain control through a combination of peripheral and central mechanisms. Peripheral mechanisms include spinal gate control modulation and substance P decrease, whereas central mechanisms include endogenous opioid release. In the peripheral tissues, HILT reduces the release of bradykinin and histamine, increasing the pain perception threshold.

The researchers aimed to underline the advantages of HILT in musculoskeletal pathology; the present paper focuses on knee osteoarthritis as a major cause of disability in adult and old population. 

Before and after trials showed the HILT is associated with pain reduction after the first session, with continuing improvement after up to 10 sessions, and lingering results after 1 to 3 weeks. Compared to placebo/sham, HILT had significant better pain and functional improvement at the end of the therapy that was maintained for up to 3 months. However, one trial failed to document any superiority of HILT over placebo regarding the functional status of the patients. HILT and LLLT were able to achieve significant pain control and functional improvement at the completion of the therapy, lasting for at least one month. HILT was found to have better values than LLLT at all moments. Adding HILT to the conservative management of KO (i.e., physical exercise, different combinations of physical agents, oral glucosamine/chondroitin supplements) was documented to improve pain and functional status and reduce disability significantly. Ultrasound examination of the osteoarthritic knee revealed a reduction in joint effusion and synovial thickness. Any evolution of cartilage thickness is questionable as a small number of trials carried out controversial results. 

Doses and timing of HILT applications varied throughout the published papers. Although analgesic action may be visible after a small number of sessions (one to three), most researchers agree on an amount of eight to twelve sessions, either daily or with a frequency of two–three/week. Doses per session are mostly 300 J for analgesic effect and 3000 J for biostimulatory action. 

Research has documented the immediate effect of HILT on pain and function. Some papers followed patients for variable timeframes after therapy and reported the maintenance of results after 1 week to 4 months. 

This paper underlines the immediate, cumulative, and long-lasting analgesic effect of HILT. Function and disability are a direct consequence of pain, but other factors, such as inflammation and contractures, also contribute. Due to deep thermal action, HILT may act on those factors affecting functioning and disability, producing significant improvement comparative to other physical agents. 

Pain, functioning, and disability approaches in KO are based on pharmacological and non-pharmacological modalities; this last category includes patient education and life-style modification, physical exercise, and a combination of physical agents, with HILT among them, with important analgesic and anti-inflammatory effects. 

HILT was used in the management of different shoulder pathologies, such as subacromial impingement syndrome [42], frozen shoulder [43,44], chronic cervical pain [8], and lower back pain [45].

High-intensity laser therapy produced significant improvements in pain and may improve the functioning and disability of patients with knee osteoarthritis. HILT must be a part of the therapeutic approach to KO, functioning as a non-pharmacologic tool with which to increase the effectiveness of associated modalities. 

## Figures and Tables

**Figure 1 biomedicines-12-01679-f001:**
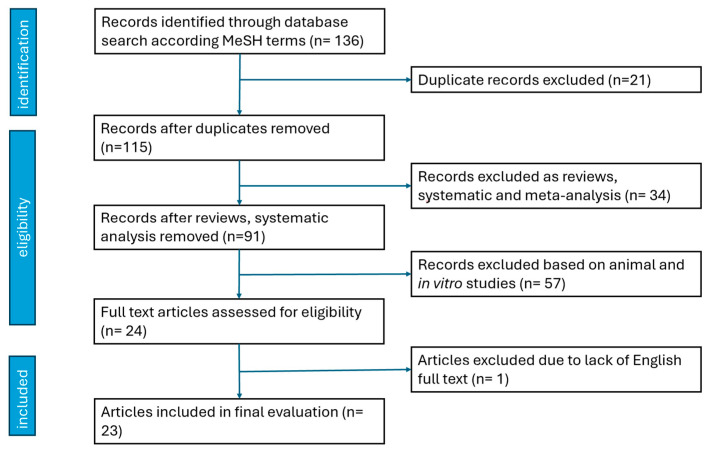
Selection process of relevant trials.

## Data Availability

All data are available on internet resources.

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
