# Peer review of "High-Intensity Laser Therapy in Pain Management of Knee Osteoarthritis"

_biomedicines, 2024, doi:10.3390/biomedicines12081679_

Round 1

Reviewer 1 Report (Previous Reviewer 1)

Comments and Suggestions for Authors

Dear Authors,

Thank you for revising the manuscript and reflecting the reviewers' comments. I am happy with the revised version of the manuscript. 

I only have one comment for you:

As laser therapy is one of the physical stimulation techniques, I suggest adding the following keywords to the manuscript: "physical stimulation"

Comments on the Quality of English Language

I suggest performing an English language editing/proofreading before publication

Author Response

Thank you for your attention in this paper.

I added the keywords „physical stimulation”.

I hope the paper fulfills the requirements.

Reviewer 2 Report (Previous Reviewer 2)

Comments and Suggestions for Authors

Thank you for your response and for resubmitting the paper.

I decided to address a few comments. I regret that this is not a systematic review, as it would significantly improve the quality and scientific value of the work.

Regarding the lack of meta-analysis due to the heterogeneity of the studies, this can be determined based on appropriate tests that were not conducted. So again, it is a pity that the authors did not perform a meta-analysis.

However, these are not critical errors that determine the value of the paper. Therefore, I am allowing the paper to proceed. Nevertheless, I request that you correct the response to question 10. Justifying the subjective opinion about the inclusion of high-quality literature in the databases may not be sufficient.

I will help the authors and ask them to cite other reviews that are based on these databases as evidence of their use in reviews. I suggest the following example papers: 10.3390/ijerph192012994; 10.3390/jcm13051365; 10.3390/healthcare12121234.

Best regards

Author Response

Thank you for your commitement in this paper. 

Regarding the question 10, we consider valuable the databases we searched on an objective opinion, as an important number of review papers are based on them. Furthermore, we agree that traditional approaches of musculoskeletal pathologies are increasingly studied and successfully applied. We studied the suggested references and added the suitable ones in the paper, as they include valuable information.

This manuscript is a resubmission of an earlier submission. The following is a list of the peer review reports and author responses from that submission.

Round 1

Reviewer 1 Report

Comments and Suggestions for Authors

Dear Authors,

The manuscript reviews the effects of high-intensity laser therapy in pain management of knee osteoarthritis. The scope of this review paper is important and the evaluations performed in the manuscript are useful.

I think that the quality of the manuscript needs to be improved before publication. There are some concerns and comments for your consideration. Please see below.

1. This is a review paper on "high-intensity laser therapy in pain management of knee osteoarthritis". You are reviewing a physical stimulation technique recently used in the pain management of knee osteoarthritis. Therefore, your paper should cover all aspects and specifications of high-intensity laser therapy method while introducing protocols and standards used in previous works. This information needs to be included in the literature review section.

2. After the introduction a literature review is required.

A literature review surveys scholarly sources and published papers on this topic. It provides an overview of current knowledge, allowing readers to identify relevant theories, methods, and gaps in the existing research.

The authors should comprehensively introduce the HILT technique that has been used in the pain management of knee osteoarthritis. The performed studies and the existing methods should be presented so that the reader can have a clear and correct understanding of the effects of this method concluded in each published work. I strongly suggest that authors work on the literature review and comprehensively review the existing methods/techniques of HILT for pain management of knee osteoarthritis. it is not sufficient to only include a table in the results section and include the results of the published works. Authors should present the methods and published works in the literature review, that are included in Table 1.

3. Line 42-43: "A few studies conducted on osteoarthritis, particularly on knee osteoarthritis (KO) raised attention to this topic." These studies should be included in the literature review. 

4. Sections 2 and 3 look good to me. 

5. Section 4 Discussion: please elaborate on subsections 4.1, 4.2., and 4.3 and include more details. for example: "As for intergroup analysis, all trials but one reported significantly better results in the HILT group."

What are the evaluation factors that show better results when HILT is used? please include them. The presented results should be compared so that the claim can be justified. 

Comments on the Quality of English Language

language editing and proofreading required.

Author Response

Comments 1. This is a review paper on "high-intensity laser therapy in pain management of knee osteoarthritis". You are reviewing a physical stimulation technique recently used in the pain management of knee osteoarthritis. Therefore, your paper should cover all aspects and specifications of high-intensity laser therapy method while introducing protocols and standards used in previous works. This information needs to be included in the literature review section.

Response 1. Thank you for your observation. We introduced information about this new technique and her therapeutical use.

Comments 2. After the introduction a literature review is required.

A literature review surveys scholarly sources and published papers on this topic. It provides an overview of current knowledge, allowing readers to identify relevant theories, methods, and gaps in the existing research.

The authors should comprehensively introduce the HILT technique that has been used in the pain management of knee osteoarthritis. The performed studies and the existing methods should be presented so that the reader can have a clear and correct understanding of the effects of this method concluded in each published work. I strongly suggest that authors work on the literature review and comprehensively review the existing methods/techniques of HILT for pain management of knee osteoarthritis. it is not sufficient to only include a table in the results section and include the results of the published works. Authors should present the methods and published works in the literature review, that are included in Table 1.

Response 2. Thank you for your observation. In the Introduction section we introduced a literature review on different utilization of HILT in musculoskeletal pain, as the main indication of this therapy.

Comments 3. Line 42-43: "A few studies conducted on osteoarthritis, particularly on knee osteoarthritis (KO) raised attention to this topic." These studies should be included in the literature review. 

Response 3. Thank you for your observation. We included these studies in the review.

Comments 4. Sections 2 and 3 look good to me. 

Response 4. Thank you.

Comments 5. Section 4 Discussion: please elaborate on subsections 4.1, 4.2., and 4.3 and include more details. for example: "As for intergroup analysis, all trials but one reported significantly better results in the HILT group."

Response 5. Thank you for your observation. We elaborated more on every section, we added more details on evaluation scales and results, also on long term observation for the specified trials.

Comments 6. What are the evaluation factors that show better results when HILT is used? please include them. The presented results should be compared so that the claim can be justified. 

Response 6. Thank you for your observation. In the Conclusion section we summarized the results of our paper.

Finally, we hope that we will fulfill the recommendations to write a high quality material.

Reviewer 2 Report

Comments and Suggestions for Authors

Thank you for submitting the paper for review. I have the following questions:

1.      Why did the authors not decide to conduct a systematic review?

2.      Why did the authors not register this review anywhere?

3.      Why did the authors not choose to conduct a meta-analysis, which would have been significantly more valuable?

4.      Additionally, what period was analyzed?

5.      When was the literature review started and when was it completed?

6.      Identify the authors who reviewed the literature.

7.      Provide information on how disputes among the authors were resolved. Were there any?

8.      Figure 1 – correct according to PRISMA standards.

9.      Additionally, describe the inclusion and exclusion criteria according to PICO standards.

10.   'PubMed, Google, Scholar, Web of Science' – why were these databases chosen? Please justify.

11.   The introduction in the paper is too short, lacking justification for conducting the study in a health and economic context. Epidemiological data are missing.

12.   An additional note: at the end of the sentence, the citation should be placed before the period, not the other way around.

At this point, I am suspending further review until the above issues are addressed.

Author Response

Comments 1. Why did the authors not decide to conduct a systematic review?

Response 1. Certainly, a systematic review is an important tool to assess the quality of research. HILT is a new physical therapy, with a small number of patients included in different types of trials, with a high variability of therapeutical protocols. Even if the first mentioned trial in literature dates from 2009, only beginning with 2016 there were some more papers published, with a rhythm of 2 papers per year (2016, 2017, 2018) and in some years no published paper. Lately, in 2022 and 2023 the literature became somewhat richer (9 trials). Additionally, there is heterogeneity of the study groups between the trials. These are the arguments for not conducting a systematic review and rather a descriptive one.

Comments 2. Why did the authors not register this review anywhere?

Response 2. As we did not conduct a systematic review, we did not register it.

Comments 3. Why did the authors not choose to conduct a meta-analysis, which would have been significantly more valuable?

Response 3. A meta-analysis is indeed more valuable. In our paper on HILT on knee osteoarthritis, we consider that clinical heterogeneity and clinical factors may affect the impact of HILT on the outcomes.

Comments 4. Additionally, what period was analyzed?

Response 4. Thank you for your observation. We added the timeframe of our research in the text.

Comments 5. When was the literature review started and when was it completed?

Response 5. Thank you for your observation. We revised the text, adding start and the completion of the literature review.

Comments 6. Identify the authors who reviewed the literature.

Response 6. Thank you for your observation. We reformulated the Material and Method section according to your recommendation.

Comments 7. Provide information on how disputes among the authors were resolved. Were there any?

Response 7. Thank you for your observation. There were some disputes among the authors, we introduced the topic in the paper.

Comments 8. Figure 1 – correct according to PRISMA standards.

Response 8. Thank you for your observation. We corrected the Figure 1 accordingly.

Comments 9. Additionally, describe the inclusion and exclusion criteria according to PICO standards.

Response 9. Thank you for your observation. We modified the section and added the PICO inclusion criteria.

Comments 10. 'PubMed, Google, Scholar, Web of Science' – why were these databases chosen? Please justify.

Response 10. Thank you for your question. We chose these databases as they are extensive and contain high quality literature.

Comments 11. The introduction in the paper is too short, lacking justification for conducting the study in a health and economic context. Epidemiological data are missing.

Response 11. Thank you for your observation. We added in the introduction section more information to fulfill the request.

Comments 12.   An additional note: at the end of the sentence, the citation should be placed before the period, not the other way around.

Response 12. Thank you for your observation. We performed the modifications accordingly.

We hope that we will fulfill the recommendations to write a high quality material.